# A polar prediction model for learning to represent visual transformations

**Pierre-Étienne H. Fiquet**[1]     **Eero P. Simoncelli**[1,2]

[1] Center for Neural Science, New York University
[2] Center for Computational Neuroscience, Flatiron Institute
`{pef246, eero.simoncelli}@nyu.edu`

## Abstract

All organisms make temporal predictions, and their evolutionary fitness level depends on the accuracy of these predictions. In the context of visual perception, the motions of both the observer and objects in the scene structure the dynamics of sensory signals, allowing for partial prediction of future signals based on past ones. Here, we propose a self-supervised representation-learning framework that extracts and exploits the regularities of natural videos to compute accurate predictions. We motivate the polar architecture by appealing to the Fourier shift theorem and its group-theoretic generalization, and we optimize its parameters on next-frame prediction. Through controlled experiments, we demonstrate that this approach can discover the representation of simple transformation groups acting in data. When trained on natural video datasets, our framework achieves better prediction performance than traditional motion compensation and rivals conventional deep networks, while maintaining interpretability and speed. Furthermore, the polar computations can be restructured into components resembling normalized simple and direction-selective complex cell models of primate V1 neurons. Thus, polar prediction offers a principled framework for understanding how the visual system represents sensory inputs in a form that simplifies temporal prediction.

## 1 Introduction

The fundamental problem of vision can be framed as that of representing images in a form that is more useful for performing visual tasks, be they estimation, recognition, or motor action. Perhaps the most general "task" is that of temporal prediction, which has been proposed as a fundamental goal for unsupervised learning of visual representations [1]. Previous research along these lines has generally focused on estimating stable representations rather than using them to predict: for example, extracting slow features [2], or finding sparse codes that have slow amplitudes and phases [3].

In video processing and computer vision, a common strategy for temporal prediction is to first estimate local translational motion, and then (assuming no acceleration) use this to warp and/or copy previous content to predict the next frame. Such motion compensation is a fundamental component in video compression schemes as MPEG [4]. These video coding standards are the result of decades of engineering effort [5], and have enabled reliable and efficient digital video communication that is now commonplace. But motion estimation is a difficult nonlinear problem, and existing methods fail in regions where temporal evolution is not translational and smooth: for example, expanding or rotating motions, discontinuous motion at occlusion boundaries, or mixtures of motion arising from semi-transparent surfaces (e.g., viewing the world through a dirty pane of glass). In compression schemes, these failures of motion estimation lead to prediction errors, which must then be adjusted by sending additional corrective bits.

37th Conference on Neural Information Processing Systems (NeurIPS 2023).

Human perception does not seem to suffer from such failures—subjectively, we can anticipate the time-evolution of visual input even in the vicinity of these commonly occurring non-translational changes. In fact, those changes are often highly informative, revealing object boundaries, and providing ordinal depth and shape cues and other information about the visual scene. This suggests that the human visual system uses a different strategy, perhaps bypassing altogether the explicit estimation of local motion, to represent and predict evolving visual input. Toward this end, and inspired by the recent hypothesis stating that primate visual representations support prediction by "straightening" the temporal trajectories of naturally-occurring input [6], we formulate an objective for learning an image representation that facilitates prediction by linearizing the temporal trajectories of frames in natural videos.

To motivate the separation of spatial representation and temporal prediction, we first consider the special case of rigidly translating signals in one dimension. In the frequency domain, translation corresponds to phase advance (section 2.1), which reduces prediction to phase extrapolation (section 2.2). We invoke basic arguments from group theory to motivate the search for generalized representations (section 2.3). We propose a neural network architecture that maps individual video frames to a latent space where prediction can be computed more readily and then mapped back to generate an estimated frame (section 3). We train the entire system end-to-end to minimize next frame prediction errors and verify that, in controlled experiments, the learned weights recover the representation of the group used to generate synthetic data. On natural video datasets, our framework consistently outperforms conventional motion compensation methods and is competitive with deep predictive neural networks (section 4). We establish connections between each element of our framework and the modeling of early visual processing (section 4.3).

## 2 Background

### 2.1 Base case: the Fourier shift theorem

Our approach is motivated by the well-known behavior of Fourier representations with respect to signal translation. Specifically, the complex exponentials that constitute the Fourier basis are the eigenfunctions of the translation operator, and translation of inputs produces systematic phase advances of frequency coefficients. Let $\mathbf{x} = [x_0, \ldots, x_{N-1}]^\top \in \mathbb{R}^N$ be a discrete signal indexed by spatial location $n \in [0, N-1]$, and let $\widetilde{\mathbf{x}} \in \mathbb{C}^N$ be its Fourier transform indexed by $k \in [0, N-1]$. We write $\mathbf{x}^{\downarrow v}$, the circular shift of $\mathbf{x}$ by $v$, $x_n^{\downarrow v} = x_{n-v}$ (with translation modulo $N$). Let $\phi$ denote the primitive $N^\text{th}$ root of unity, $\phi = e^{i2\pi/N}$, and let $\mathbf{F} \in \mathbb{C}^{N \times N}$ denote the Fourier matrix, $F_{nk} = \frac{1}{\sqrt{N}}\phi^{nk}$. Multiplication by the adjoint (i.e. the conjugate transpose), $\mathbf{F}^*$, gives the Discrete Fourier Transform (DFT), and by $\mathbf{F}$ the inverse DFT. We can express the Fourier shift theorem[1] as: $\mathbf{x}^{\downarrow v} = \mathbf{F}\text{diag}(\boldsymbol{\phi}_v)\mathbf{F}^*\mathbf{x}$, where $\boldsymbol{\phi}_v = [\phi^0, \phi^{-v}, \phi^{-2v}, \ldots, \phi^{-(n-1)v}]^\top$ is the vector of phase shift by $v$. See Appx. A for a consolidated list of notation used throughout this work.

This relationship can be depicted in a compact diagram:

$$
\begin{array}{ccc}
\widetilde{x}_k & \xrightarrow{\text{phase shift}} & \phi^{-kv}\widetilde{x}_k \\
\uparrow{\scriptstyle \mathbf{F}^*} & & \downarrow{\scriptstyle \mathbf{F}} \\
x_n & \xrightarrow{\text{spatial shift}} & x_{n-v}
\end{array}
\tag{1}
$$

This diagram illustrates how transforming to the frequency domain renders translation a "simpler" operation: phase shift acts as rotation in each frequency subspace, i.e. it is diagonalized.

---

[1] Proof by substituting $m = n - v$:

$$
\widetilde{x^{\downarrow v}}_k = \sum_{n=0}^{N-1} \phi^{-kn} x_{n-v} = \sum_{m=-v}^{N-1-v} \phi^{-kv}\phi^{-km} x_m = \phi^{-kv} \sum_{n=0}^{N-1} \phi^{-kn} x_n = \phi^{-kv}\widetilde{x}_k.
$$

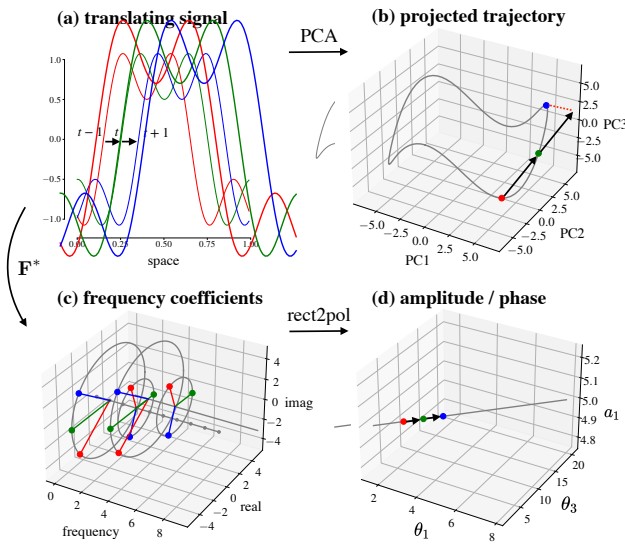

**Figure 1: Straightening translations.** **(a)** Three snapshots of a translating signal consisting of two superimposed sinusoidal components: $x_{n,t} = \sin(2\pi(n - t)) + \sin(2\pi 3(n - t))/2$. **(b)** Projection of the signal into the space of the top three principal components. The colored points correspond to the three snapshots in panel (a). In signal space, the temporal trajectory is highly curved—linear extrapolation fails. **(c)** Complex-valued Fourier coefficients of the signal as a function of frequency. The temporal trajectory of the frequency representation is the phase advance of each sinusoidal component. **(d)** Trajectory of one amplitude and both (unwrapped) phases components. The conversion from rectangular to polar coordinates reduces the trajectory to a straight line—which is predictable via linear extrapolation.

## 2.2 Prediction via phase extrapolation

Consider a signal, the $N$-dimensional vector $\mathbf{x}_t$, that translates at a constant velocity $v$ over time: $x_{n,t} = x_{n-vt,0}$. This sequence traces a highly non-linear trajectory in signal space, i.e. the vector space where each dimension corresponds to the signal value at one location. In this space, linear extrapolation fails. As an example, Figure 1 shows a signal consisting of a sum of two sinusoidal components in one spatial dimension. Mapping the signal to the frequency domain simplifies the description. In particular, the translational motion now corresponds to circular motion of the two (complex-valued) Fourier coefficients associated with the constituent sinusoids. In polar coordinates, the trajectory of these coefficients is straight, with both phases advancing linearly (at a rate proportional to their frequency), and both amplitudes constant.

## 2.3 Generalization: representing transformation groups

Streaming visual signals are replete with structured transformations, such as object displacements and surface deformations. While these can not be captured by the Fourier representation, which only handles global translation, the concept of representing transformations in their eigen-basis generalizes. Indeed, representation theory describes elements of general groups as linear transformations in vector spaces, and decomposes them into basic building blocks [7]. However, the transformation groups acting in image sequences are not known a priori, and it can be difficult to give an explicit representation of general group actions. In this work, we aim to find structures that can be modeled as groups in image sequences, and we learn their corresponding representations from unlabeled data.

In harmonic analysis, the Peter-Weyl Theorem (1927) establishes the completeness of the unitary irreducible representations for compact topological groups (an irreducible representation is a subspace that is invariant to group action and that can not be further decomposed). Furthermore, every compact Lie group admits a faithful (i.e. injective) representation given by an explicit complete orthogonal basis, constructed from finite-dimensional irreducible representations [7]. Accordingly, the action of a compact Lie group can be expressed as a rotation within each irreducible representation (an example is the construction of steerable filters [8] in the computational vision literature).

In the special case of compact commutative Lie groups, the irreducible representations are one-dimensional and complex-valued (alternatively, pairs of real valued basis functions). These groups have a toroidal topology and, in this representation, their action can be described as advances of the phases. This suggests a strategy for learning a representation: seek pairs of basis functions for which phase extrapolation yields accurate prediction of upcoming images in a stream of visual signals.

# 3 Methods

## 3.1 Objective function

We aim to learn a representation of video frames that enables next frame prediction. Specifically, we optimize a cascade of three parameterized mappings: an analysis transform ($f_w$) that maps each frame to a latent representation, a prediction in the latent space ($p_w$), and a synthesis transform ($g_w$) that maps the predicted latent values back to the image domain. The parameters $w$ of these mapping are learned by minimizing the average squared prediction error:

$$\min_w \sum_t \|\mathbf{x}_{t+1} - g_w(\hat{\mathbf{z}}_{t+1})\|^2; \qquad \text{where } \hat{\mathbf{z}}_{t+1} = p_w(\mathbf{z}_t, \mathbf{z}_{t-1}), \text{and } \mathbf{z}_t = f_w(\mathbf{x}_t). \qquad (2)$$

An instantiation of this framework is illustrated in Figure 2. Here the analysis and synthesis transforms are adjoint linear operators, and the predictor is a diagonal phase extrapolation.

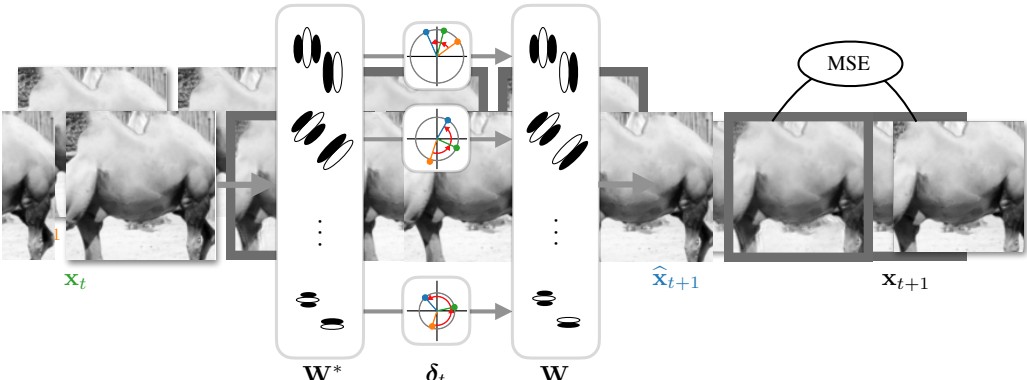

**Figure 2: Polar prediction model**. The previous and current images in a sequence ($\mathbf{x}_{t-1}$ and $\mathbf{x}_t$) are convolved with pairs of filters ($\mathbf{W}^*$), each yielding complex-valued coefficients. For a given spatial location in the image, the coefficients for each pair of filters are depicted in complex planes with colors corresponding to time step. The coefficients at time $t + 1$ are predicted from those at times $t - 1$ and $t$ by extrapolating the phase ($\boldsymbol{\delta}_t$). These predicted coefficients are then convolved with the adjoint filters ($\mathbf{W}$) to generate a prediction of the next image in the sequence ($\hat{\mathbf{x}}_{t+1}$). This prediction is compared to the next frame ($\mathbf{x}_{t+1}$) by computing the mean squared error (MSE) and the filters are learned by minimizing this error. Notice that, at coarser scales, the coefficient amplitudes tend to be larger and the phase advance smaller, compared to finer scales.

## 3.2 Analysis-synthesis transforms

**Local processing**   When focusing on a small spatial region in an image sequence, the transformation observed as time passes can often be well approximated as a *local* translation. That is to say, in a spatial neighborhood around position $n, m \in N(n)$, we have: $x_{m,t+1} \approx x_{m-v,t}$. We can use the decomposition described for global rigid translation, replacing the Fourier transform with a learned local convolutional operator [9], processing each spatial neighborhood of the image independently and in parallel.

At every position in the image (spatial indices are omitted for clarity of notation), each pair of coefficients is computed as an inner product between the input and the filter weights of each pair of channels. Specifically for $k \in [0, K]$, where $K$ is the number of pairs of channels, we have: $y_{2k,t} = \mathbf{v}_{2k}^\top \mathbf{x}_t$ and $y_{2k+1,t} = \mathbf{v}_{2k+1}^\top \mathbf{x}_t$. Correspondingly, an estimated next frame is generated by applying the transposed convolution $g_w$ to advanced coefficients (see section 3.3). We use the same weights for the encoding and decoding stages, that is to say the analysis operator is the conjugate transpose of the synthesis operator (as is the case for for the Fourier transform and its inverse). Sharing these weights reduces the number of parameters and simplifies interpretation of the learned solution.

**Multiscale processing**  Transformations such as translation act on spatial neighborhoods of various sizes. To account for this, the image is first expanded at multiple resolutions in a fixed overcomplete Laplacian pyramid [10]; then learned spatial filtering (see previous paragraph) and temporal processing (see section 3.3) are applied on these coefficients; and finally, the modified coefficients are recombined across scales to generate the predicted next frame (see details in the caption of Figure 3).

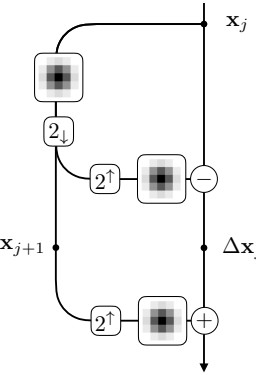

**Figure 3: Laplacian pyramid.** An image is recursively split into low frequency approximation and high frequency details. Given the initial image $\mathbf{x} = \mathbf{x}_{j=0} \in \mathbb{R}^N$, the low frequency approximation (aka. Gaussian pyramid coefficients) is computed via blurring (convolution with a fixed filter $B$) and downsampling ("stride" of 2, denoted $2_\downarrow$): $\mathbf{x}_j = 2_\downarrow(B \star \mathbf{x}_{j-1} \in \mathbb{R}^{2^{-j}N})$, for levels $j \in [1, J]$; and the high frequency details (aka. Laplacian pyramid coefficients) are computed via upsampling (put one zero between each sample, $2^\uparrow$) and blurring: $\Delta \mathbf{x}_j = \mathbf{x}_j - B \star (2^\uparrow \mathbf{x}_{j+1})$. These coefficients, $\{\Delta \mathbf{x}_j\}_{0 \leq j < J}$, as well as the lowpass, $x_J$, can then be further processed. A new image is constructed recursively on these processed coefficients. First by upsampling the lowest resolution, and then by adding the corresponding details until the initial scale $j = 0$ as: $\mathbf{x}_j = B \star (2^\uparrow \mathbf{x}_{j+1}) + \Delta \mathbf{x}_j$.

## 3.3 Prediction mechanism

**Polar predictor**  In order to obtain phases, we group coefficients in pairs, express them as complex-valued: $z_{k,t} = y_{2k,t} + iy_{2k+1,t} \in \mathbb{C}$, and convert to polar coordinates: $z_{k,t} = a_{k,t}e^{i\theta_{k,t}}$. With this notation, linear phase extrapolation amounts to $\hat{z}_{k,t+1} = a_{k,t}e^{i(\theta_{k,t} + \Delta\theta_{k,t})}$, where the phase advance $\Delta\theta_{k,t}$ is equal to the phase difference over the interval from $t-1$ to $t$: $\Delta\theta_{k,t} = \theta_{k,t} - \theta_{k,t-1}$. Note that we assume no phase acceleration and constant amplitudes. The phase-advanced coefficients can be expressed in a more direct way, using complex arithmetic, as:

$$\hat{z}_{k,t+1} = \delta_{k,t} z_{k,t}, \text{ where } \delta_{k,t} = \frac{z_{k,t}\overline{z_{k,t-1}}}{|z_{k,t}||z_{k,t-1}|}, \tag{3}$$

with $\overline{z}$ and $|z|$ respectively denoting complex conjugation and complex modulus of $z$. This formulation in terms of products of complex coefficients[2] has the benefit of handling phases implicitly, which makes phase processing computationally feasible, as previously noted in the texture modeling literature [11, 12]. Optimization over circular variables suffers from a discontinuity if one represents the variable over a finite interval (e.g. $[-\pi, \pi]$). Alternatively, procedures for "unwrapping" the phase are generally unstable and sensitive to noise.

In summary, given a video dataset $X = [\mathbf{x}_1, \ldots, \mathbf{x}_T] \in \mathbb{R}^{N \times T}$, the convolutional filters of a polar prediction model are learned by minimizing the average squared prediction error:

$$\min_{\mathbf{W} \in \mathbb{C}^{N \times NK}} \sum_{t=1}^{T} ||\mathbf{x}_{t+1} - \mathbf{W}\text{diag}(\boldsymbol{\delta}_t)\mathbf{W}^*\mathbf{x}_t||^2;$$
$$\text{where } \boldsymbol{\delta}_t = (\mathbf{z}_t \odot \overline{\mathbf{z}_{t-1}}) \oslash (|\mathbf{z}_t| \odot |\mathbf{z}_{t-1}|), \text{ and } \mathbf{z}_t = \mathbf{W}^*\mathbf{x}_t. \tag{4}$$

The columns of the convolutional matrix $\mathbf{W}$ contain the $K$ complex-valued filters, $\mathbf{w}_k = \mathbf{v}_{2k} + i\mathbf{v}_{2k+1} \in \mathbb{C}^N$ (repeated at $N$ locations) and multiplication, division, amplitude, and complex conjugation are computed pointwise.

This "polar predictor" (hereafter, **PP**) is depicted in figure 2. Note that the conversion to polar coordinates is the only non-linear step used in the architecture. This bivariate non-linear activation function differs markedly from the typical (pointwise) rectification operations found in convolutional neural networks. Note that the polar predictor is homogeneous of degree one, since it is computed as the ratio of a cubic over a quadratic.

---

[2]Using "Gauss's trick", each complex multiplication can be computed with only three real multiplications: $(a + ib)(c + id) = ac - bd + i((a + b)(c + d) - ac - bd)$.

**Quadratic predictor**    Rather than building in the phase extrapolation mechanism, we now consider a more expressive parametrization of the predictor ($p_w$) that can be learned on data jointly with the analysis and synthesis mappings. This "quadratic predictor" (hereafter, **QP**) generalizes the polar extrapolation mechanism and can accommodate groups of channels of size larger than two (as for the real and imaginary part in the polar predictor).

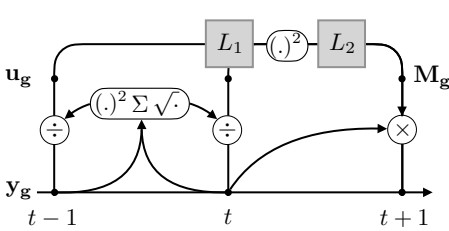

**Figure 4: Learnable quadratic prediction mechanism.** Groups of coefficients ($\mathbf{y}_{k,t}$) at the previous and current time-step are normalized ($\mathbf{u}_{k,t}$) and then passed through in a Linear-Square-Linear cascade to produce a prediction matrix ($\mathbf{M}_{k,t}$). This matrix is applied to the current vector of coefficients to predict the next one. The linear transforms ($\mathbf{L}_1$ and $\mathbf{L}_2$) are learned. This quadratic prediction module contains phase extrapolation as a special case and handles the more general case of groups of coefficients beyond pairs.

First, we rewrite the phase extrapolation mechanism of equation 3 using only real-valued elements:

$$\begin{bmatrix} \hat{y}_{2k,t+1} \\ \hat{y}_{2k+1,t+1} \end{bmatrix} = \begin{bmatrix} \cos \Delta\theta_{k,t} & -\sin \Delta\theta_{k,t} \\ \sin \Delta\theta_{k,t} & \cos \Delta\theta_{k,t} \end{bmatrix} \begin{bmatrix} y_{2k,t} \\ y_{2k+1,t} \end{bmatrix}, \tag{5}$$

where the $2 \times 2$ prediction matrix, $\mathbf{M}_{k,t}$, is a rotation by angle $\Delta\theta_{k,t}$. Using an elementary trigonometric identity: $\cos(a - b) = \cos(a)\cos(b) + \sin(a)\sin(b)$ and recalling that $y_{k,t} = \mathbf{v}_{2k}^\top \mathbf{x}_t = a_{k,t}\cos(\theta_{k,t})$, the elements of this prediction matrix can be made explicit. They are a quadratic function of the normalized response, which we write $u_{2k,t}$ (for unit vector), as:

$$\cos \Delta\theta_{k,t} = u_{2k,t}u_{2k,t-1} + u_{2k+1,t}u_{2k+1,t-1}, \text{ where } u_{2k,t} = \frac{y_{2k,t}}{(y_{2k,t}^2 + y_{2k+1,t}^2)^{1/2}}. \tag{6}$$

This quadratic function can be expressed in terms of squared quantities (without cross terms) using the polarisation identity: $\alpha\beta = ((\alpha + \beta)^2 - (\alpha - \beta)^2)/4$. Specifically:

$$\cos \Delta\theta_{k,t} = \frac{1}{4}\Big((u_{2k,t} + u_{2k,t-1})^2 - (u_{2k,t} - u_{2k,t-1})^2 +$$
$$(u_{2k+1,t} + u_{2k+1,t-1})^2 - (u_{2k+1,t} - u_{2k+1,t-1})^2\Big). \tag{7}$$

An analogous expression can be derived for $\sin \Delta\theta_{k,t}$. These operations are depicted in Figure 4: current and previous pairs of coefficients ($\mathbf{y}_{k,t} = [y_{2k,t}, y_{2k+1,t}, y_{2k,t-1}, y_{2k+1,t-1}]^\top \in \mathbb{R}^4$) are normalized ($\mathbf{u}_{k,t} \in \mathbb{R}^4$), then linearly combined ($\mathbf{L_1} \in \mathbb{R}^{4 \times d}$), pointwise squared, and linearly combined again ($\mathbf{L_2} \in \mathbb{R}^{4 \times d}$) to produce a prediction matrix ($\mathbf{M}_{k,t} \in \mathbb{R}^{2 \times 2}$) that is applied to the current coefficients to produce a prediction ($[\hat{y}_{2k,t+1}, \hat{y}_{2k+1,t+1}]^\top \in \mathbb{R}^2$). These linear combination can be learned jointly with the analysis and synthesis weights by minimizing the prediction error.

In summary, the quadratic predictor is learned as:

$$\min_{\mathbf{V},\mathbf{L}_1,\mathbf{L}_2} \sum_{t=1}^{T} ||\mathbf{x}_{t+1} - \mathbf{V}\mathbf{\Lambda}_t \mathbf{V}^\top \mathbf{x}_t||^2; \quad \text{where } \mathbf{\Lambda}_t = \text{blockdiag}(\mathbf{M}_{1,t}, \dots, \mathbf{M}_{K,t}),$$
$$\mathbf{M}_{k,t} = \mathbf{L}_2(\mathbf{L}_1^\top \mathbf{u}_{k,t})^{\odot 2}, \quad \mathbf{u}_{k,t} = \frac{\mathbf{y}_{k,t}}{(\mathbf{1}_g^\top \mathbf{y}_{k,t}^2)^{1/2}}, \quad \text{and} \quad \mathbf{y}_t = \mathbf{V}^\top \mathbf{x}_t. \tag{8}$$

The columns of the convolutional matrix $\mathbf{V} \in \mathbb{R}^{N \times gNK}$ contain the groups of $g$ filters (repeated at N locations). The number of channels in a group is no longer limited to pairs ($|g| \geq 2$ is a hyper-parameter). The prediction matrices, $\mathbf{M}_{k,t} \in \mathbb{R}^{g \times g}$, are computed as a Linear-Square-Linear cascade on normalized activity from group of coefficients at the previous two time points: $\mathbf{u}_{k,t} = [u_{k,t}, u_{k+1,t}, \dots, u_{k+g-1,t}]^\top \in \mathbb{R}^g$ and $\mathbf{u}_{k,t-1}$. The linear combinations are learnable matrices $\mathbf{L}_1 \in \mathbb{R}^{2g \times d}$ and $\mathbf{L}_2 \in \mathbb{R}^{g^2 \times d}$, where $d$ is the number of quadratic units and squaring is computed pointwise. Note that in the case of pairs ($g = 2$), six quadratic units suffice ($d = 6$).

# 4 Results

## 4.1 Recovery of planted symmetries

To experimentally validate our approach, we first verified that both the PP and the QP models can robustly recover known symmetries in small synthetic datasets consisting of translating or rotating image patches. For these experiments, the analysis and synthesis transforms are applied to the entire patch (i.e., no convolution). Learned filters for each of these cases are displayed in Figure 7, Appendix B. When trained on translating image patches, the learned filters are pairs of plane waves, shifted in phase by $\pi/2$. Similarly, when trained on rotating patches, the learned filters are conjugate pairs of circular harmonics. This demonstrates that theses models can recover the (regular) representation of some simple groups from observations of signals where their transformations are acting. When the transformation is not perfectly translational (e.g. translation with open boundary condition), the learned filters are localized Fourier modes. Note that when multiple kinds of transformations are acting in data (e.g., mixtures of both translations and rotations), the PP model is forced to compromise on a single representations. Indeed, the phase extrapolation mechanism is adaptive but the basis in which it is computed is fixed and optimized. A more expressive model would also allow for adaptation of the basis itself.

## 4.2 Prediction performance on natural videos

We compare our multiscale polar predictor (**mPP**) and quadratic predictor (**mQP**) methods to a causal implementation of the traditional motion-compensated coding (**cMC**) approach. For each block in a frame, the coder searches for the most similar spatially displaced block in the previous frame, and communicate the displacement coordinates to allow prediction of frame content by translating blocks of the (already transmitted) previous frame. We also compare to phase-extrapolation within a steerable pyramid [13], an overcomplete multi-scale decomposition into oriented channels (**SPyr**). We also implemented a deep convolutional neural network predictor (**CNN**), that maps two successive observed frames to an estimate of the next frame [14]. Specifically, we use a CNN composed of 20 non-linear stages, each consisting of 64 channels, and computed with $3 \times 3$ filters without additive constants, followed by half-wave rectification. Finally, we also consider a U-net [15] which is a CNN that processes images at multiple resolutions (**Unet**). The number of non-linear stages, the number of channels and the filter size match that of the basic CNN. See descriptions in Appendix C for architectures and Appendix D for dataset and training procedures.

**Table 1: Performance comparison.** Prediction error computed on the DAVIS dataset. Values indicate mean Peak Signal to Noise Ratio (PSNR in dB) and standard deviation computed over 10 random seeds (in parentheses).

| split | Copy | cMC | SPyr | **mPP** | **mQP** | CNN | Unet |
|---|---|---|---|---|---|---|---|
| train | 21.32 | 23.83 | 25.13 | 25.31 (0.04) | 25.38 (0.11) | 25.78 (0.18) | **26.91** (0.38) |
| test | 20.02 | 22.37 | 23.82 | **24.11** (0.01) | 24.04 (0.06) | 23.58 (0.05) | 23.94 (0.06) |

Prediction results on the DAVIS dataset [16] are summarized in Table 1. First, observe that the predictive algorithms considered in this study perform significantly better than baselines obtained by simply copying the last frame, causal motion compensation, or phase extrapolation in a steerable pyramid. Second, the multiscale polar predictor performs better than the convolutional neural networks on test data (both CNN and Unet are overfit). This demonstrates the efficiency of the polar predictor: the PP model has roughly 30 times fewer parameters than the CNN and uses a single non-linearity, while the CNN and Unet contain 20 non-linear layers. Appendix E contains additional a comparison of computational costs for these algorithms: number of trainable parameters, training and inference time. Note that on this dataset, the Unet seems to overfit to the training set. Moreover, the added expressivity afforded by the multiscale quadratic predictor did not result in significant performance gains. A representative example image sequence and the corresponding predictions are displayed in Figure 5. Additional results on a second natural video dataset are detailed in Appendix E and confirm these trends.

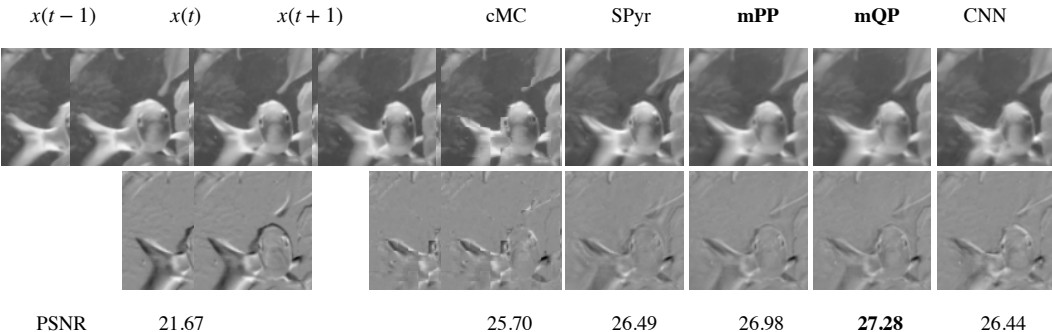

|  | $x(t-1)$ | $x(t)$ | $x(t+1)$ | cMC | SPyr | **mPP** | **mQP** | CNN |
|---|---|---|---|---|---|---|---|---|
| PSNR | | 21.67 | | | 25.70 | 26.49 | 26.98 | **27.28** | 26.44 |

**Figure 5: Example image sequence and predictions.** A typical example image sequence from the DAVIS test set. The first three frames on the top row display the unprocessed images, and last five frames show the respective prediction for each method. The bottom row displays error maps computed as the difference between the target image and each predicted next frame on the corresponding position in the first row. All subfigures are shown on the same scale.

## 4.3 Biological modeling

The polar prediction model provides a hypothesis for how cortical circuits in the primate visual system might compute predictions of their inputs ("predictive processing") [17–20]. This framework is agnostic to how predictions are used and is complementary to candidate algorithms for signaling predictions and prediction errors across the visual hierarchy ("predictive coding") [21–23].

First, the learned convolutional filters of a polar prediction model resemble receptive fields of neurons in area V1 (primary visual cortex). They are selective for orientation and spatial frequency, they tile the frequency domain (making efficient use of limited resources), and filters in each pair have similar frequency selectivity and are related by a shift in spatial phase. Representative filters are displayed in Figure 6.

Second, the quadratic prediction mechanism derived in section 3.3 suggests a qualitative bridge to physiology. Computations for this model are expressed in terms of canonical computational elements of early visual processing in primates. The normalized responses $u_{2k}(t)$ in equation 6 are linear projections of the visual input divided by the energy of related cells, similar to the normalization behavior observed in simple cells [24]. The quadratic units $m_g$ in equation 7 are sensitive to temporal change in spatial phase. This selectivity for speed in a given orientation and spatial frequency is reminiscent of direction-selective complex cells which are thought to constitute the first stage of motion estimation [25, 26].

## 5 Related work

The polar prediction model is conceptually related to representation learning methods that aim to factorize visual signals. In particular, sparse coding with complex-valued coefficients [3] aims to factorize form and motion. More generally, several methods adopt a Lie group formalism to factorize *invariance* and *equivariance*. Since the seminal work that proposed learning group generators from dynamic signals [27], a polar parametrization was explored in [28] to identify irreducible representations in a synthetic dataset, and a corresponding neural circuit was proposed in [29]. The Lie group formalism has also been combined with sparse coding [30, 31] to model natural images as points on a latent manifold. More recently, bispectral neural networks [32] have been shown to learn image representations invariant to a given global transformation. In a related formalism, factored Boltzmann machines have been proposed to learn relational features [33]. The polar prediction model differs in two important ways: (1) unlike the coding approach that operates on *iid* data, it focuses on predicting, not representing, the signal; (2) the prediction objective does not promote sparsity of either amplitude or phase components and does not rely on explicit regularization. The discontinuity arising from selection of sparse subsets of coefficients seems at odds with the representation of continuous group actions [34]. The polar prediction model relies on a smooth and continuous parameterization to jointly discover and exploit the transformations acting in sequential data. The polar prediction model is convolutional and scales to natural video data, it can adapt to the multiple unknown and noisy

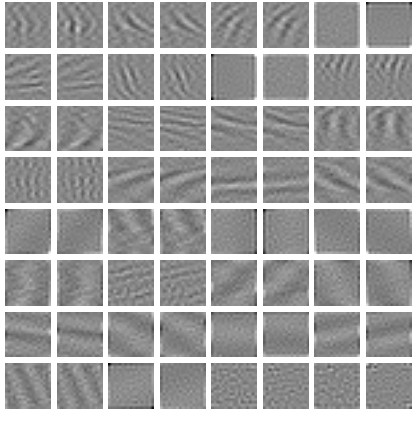

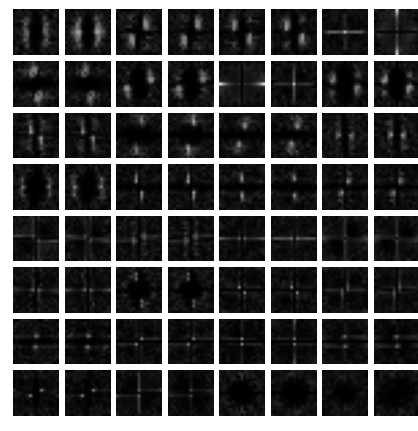

| (a) spatial domain filters | (b) corresponding Fourier amplitude spectra |

**Figure 6: Learned filters.** A single stage polar predictor was trained to predict videos from the DAVIS dataset. **(a)** Filters in each pair are displayed side by side and sorted by their norm. **(b)** Their amplitude spectra are displayed at corresponding locations. Observe that the filters are selective for orientation and spatial frequency, tile the frequency spectrum, and form quadrature pairs.

transformations that act in different spatial position. The literature on motion microscopy describes temporal processing of local phases in a fixed complex-valued wavelet representation to interpolate between video frames and magnify imperceptible movements [35]. Polar prediction processes phase in a learned representation to extrapolate to future frames.

The polar prediction model is related to other representation learning methods that also rely on temporal prediction. Temporal stability was used to learn visual representations invariant to geometric transformations occurring in image sequences [36]. The idea of learning a temporally straightened representation from image sequences was explored using a heuristic extrapolation mechanism [37] (specialized "soft max-pooling" and "soft argmax-pooling" modules). A related approach aimed at finding video representations which decompose content and pose and identify "components" in order to facilitate prediction of synthetic image sequences [38]. To tackle the challenge of natural video prediction, more sophisticated architectures have been developed for decomposing images into predictable objects [39]. A recurrent instantiation of predictive coding through time relying on a stacked convolutional LSTM architecture was proposed [40] and shown to relate to biological vision [41]. In contrast, the polar prediction model scales to prediction of natural videos while remaining interpretable. Another related approach, originating in the fluid mechanics literature, focuses on the Koopman operator [42]. This approach is a dynamical analog of the kernel trick: it lifts a system from its original state-space into a higher dimensional representation space where the dynamics can be linearized (i.e. represented by a fixed dynamics matrix). This formalism has inspired a line of work in machine learning: predictive auto-encoders learn coordinate systems that approximately linearize a system's dynamics [43]. Auxiliary networks have been introduced to adjust the dynamics matrix to velocity [44]. In contrast, the polar prediction model learns a representation that (implicitly) straightens the temporal evolution in an adaptive representation.

# 6 Discussion

We've introduced a polar prediction model and optimized it to represent visual transformations. Using adaptive phase extrapolation in a fixed shiftable basis, the model jointly discovers and exploits the approximate symmetries in image sequences which are due to local content deformation. The basis is optimized to best diagonalize video dynamics by minimizing mean squared prediction error. The phase relationships are exploited implicitly in a bundled computation, bypassing the instabilities and discontinuities of angular phase variables. By starting from mathematical fundamentals and considering an abstract formulation in terms of learning the representation of transformation groups, we formulated a framework that makes three major contributions. First, it provides a method for discovering the approximate symmetries implicit in sequential data and complements methods for imposing known invariants. Second, it achieves accurate next-frame video prediction within a

principled framework and provides an interpretable alternative to standard architectures. Third, it offers a framework to understand the nonlinear response properties of neurons in primate visual systems, potentially offering a functional explanation for perceptual straightening.

The polar prediction model makes several strong assumptions. First, it is inertial, assuming that amplitude is unchanged and phase evolves linearly with no acceleration. Second, it separates spatial and temporal processing, which seems at odds with the spatiotemporal selectivities of visual neurons [45], but could enable downstream image based tasks (e.g. object segmentation, heading direction estimation). Third, it acts independently on each coefficient, although the representation does not perfectly diagonalize the dynamics of natural videos. This is analogous to the situation in modern nonlinear signal processing where diagonal adaptive operators in appropriate bases have had a major impact in compression, denoising, and linear inverse problems [46]. Our empirical results demonstrate that these assumptions provide a reasonable description of image sequences. The standard deep networks considered here could in principle have discovered a similar solution, but they seem not to. This exemplifies a fundamental theme in computational vision and machine learning: when possible, let the representation do the analysis. An important limitation of the framework comes from the use of mean squared error, which is minimized by the posterior mean and tends to result in blurry predictions. Since temporal prediction is inherently uncertain, predictive processing should be probabilistic and exploit prior information.

The polar prediction framework suggests many interesting future directions, both for the study of visual perception (e.g. object constancy at occlusion, and object grouping from common fate) and for the development of engineering applications (e.g. building a flow-free video compression standard). To expand expressivity and better represent signal geometry, it may be possible to design a polar prediction architecture that also adapts the basis, potentially extending to the case of noncommutative transformations (e.g. the two dimensional retinal projection of three dimensional spatial rotation). Finally, it is worth considering the extension of the principles described here to prediction at longer temporal scales, which will likely require learning more abstract representations.

## Acknowledgments

We thank members of the Laboratory for Computational Vision at NYU, and of the Center for Computational Neuroscience at the Flatiron Institute for helpful discussions. This work was supported in part by the Simons Foundation.

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

# A  Notation

Let $\mathbb{R}^N$ denote the $N$-dimensional Euclidean space equipped with the usual Euclidean norm $||\cdot||$ and let $\mathbb{C}^K$ denote the $K$-dimensional complex vector space. Let $\mathbb{C}^{N\times K}$ denote the set of $N \times K$ complex-valued matrices. Let $\mathbf{F}^*$ denote the adjoint (ie. the conjugate transpose) of $\mathbf{F}$.

Given vectors $\mathbf{u}, \mathbf{v} \in \mathbb{C}^K$, let $\mathbf{u} \odot \mathbf{v} = [u_1 v_1, \dots, u_K v_K]^\top \in \mathbb{C}^K$ denote the elementwise (aka. Hadamard) product of $\mathbf{u}$ and $\mathbf{v}$. Let $\mathbf{u} \oslash \mathbf{v} = [u_1/v_1, \dots, u_K/v_K]^\top \in \mathbb{C}^K$ denote the elementwise division of $\mathbf{u}$ and $\mathbf{v}$. Let $\operatorname{diag}(\mathbf{u})$ denote the $K \times K$ diagonal matrix whose $(k, k)^{th}$ entry is $u_k$.

Let the complex number $z \in \mathbb{C}$ be expressed in rectangular coordinates as $z = x + iy$ where $(x, y) \in \mathbb{R}^2$, or in polar coordinates as $z = ae^{i\theta}$ where $(a, \theta) \in \mathbb{R}_+ \times [-\pi, \pi]$. Its complex conjugate is $\bar{z} = x - iy = ae^{-i\theta}$. The rectangular coordinates are $u = a\cos(\theta)$ and $v = a\sin(\theta)$; and the polar coordinates are $a = |z| = \sqrt{x^2 + y^2}$, and $\theta = \angle z = \operatorname{atan2}(y, x)$. We overload these notations to denote element-wise operations on a vector $\mathbf{z} \in \mathbb{C}^K$: $\bar{\mathbf{z}} = [\overline{z_1}, \dots, \overline{z_K}]^\top \in \mathbb{C}^K$ and $|\mathbf{z}| = [|z_1|, \dots, |z_K|]^\top \in \mathbb{R}_+^K$.

# B  Planted symmetries

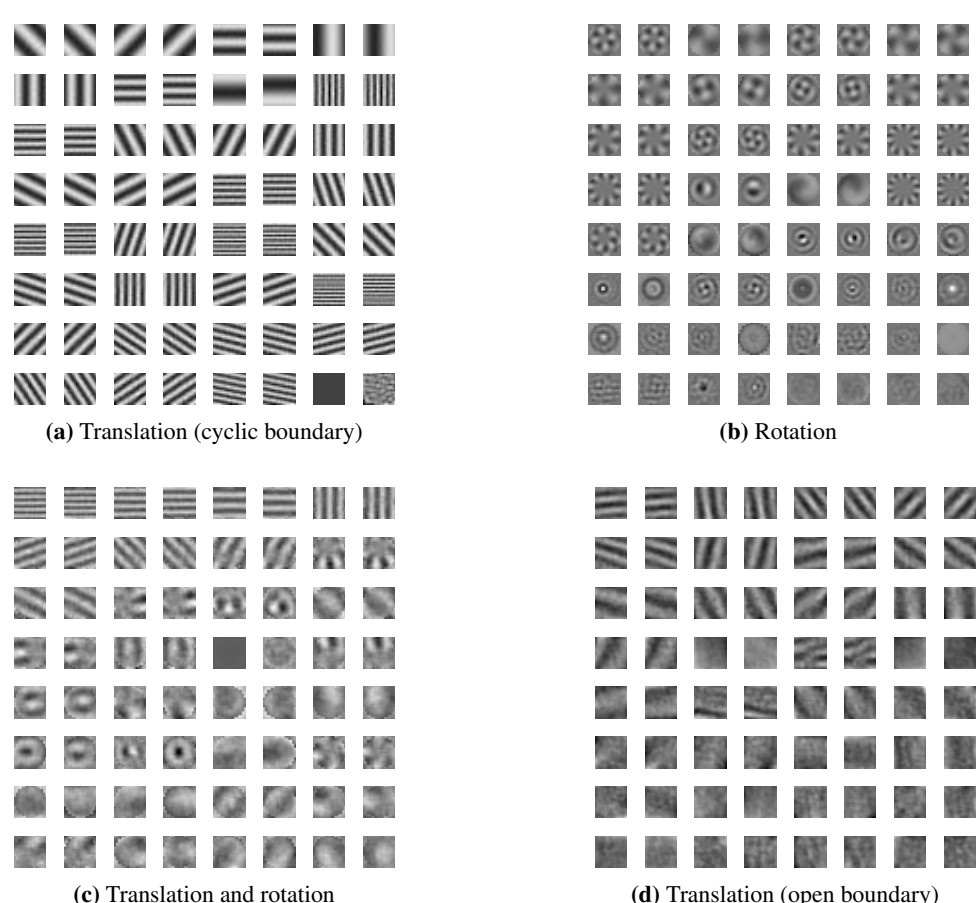

(a) Translation (cyclic boundary)  (b) Rotation

(c) Translation and rotation  (d) Translation (open boundary)

**Figure 7: Learned filters on planted symmetries.**

Filters of polar predictor networks trained to predict small synthetic sequences. We randomly select 100 image patches of size $16 \times 16$ from the DAVIS dataset and generate training data by manually transforming them—applying translations or rotations. We verify that PP recovers the corresponding harmonic functions: Fourier modes for translation (panel a), and disk harmonics for rotation (panel b). To show that the recovery of harmonics is robust, we design two additional synthetic datasets. i) the combination of translational and rotational sequences. In this case, PP learns some filters

that correspond to either transformation. But the model does not have the expressivity required to dynamically adapt the representation to either transformation (panel c); ii) generalized translation sequences: spatially sliding a square window on a large image (ie. new content creeps in and falls off at boundaries), instead of using cyclic boundary condition (ie. content wraps around the edges). In this case, PP learns localized Fourier-like modes (panel d), indicating that approximate group actions still provide meaningful training signal—but some of the filters are not structured. In each panel, the 32 pairs of filters are sorted by their norm. Notice that some of high frequency harmonics are missing. This is due to the spectral properties of the datasets, which have more power at lower frequencies.

## C  Description of architectures

**Motion Compensation**   We compare our method to the traditional motion-compensated coding approach that forms the core of inter-picture coding in well established compression standards such as MPEG. Block matching is an essential component of these standards, allowing the compression of video content by up to three orders of magnitude with moderate loss of information. For each block in a frame, typical coders search for the most similar spatially displaced block in the previous frame (typically measured with MSE), and communicate the displacement coordinates to allow prediction of frame content by translating blocks of the (already transmitted) previous frame. We implemented a "diamond search" algorithm [47] operating on blocks of $8 \times 8$ pixels, with a maximal search distance of 8 pixels which balances accuracy of motion estimates and speed of estimation (the search step is computationally intensive). We use the estimated displacements to perform causal motion compensation (**cMC**), using displacement vectors estimated from the previous two observed frames ($\mathbf{x}_{t-1}$ and $\mathbf{x}_t$) to predict the *next* frame ($\mathbf{x}_{t+1}$) rather than the current one (as in MPEG).

**Complex Steerable Pyramid**   We consider a fixed multiscale oriented representation of image content: a steerable pyramid [13, 11] covering 16 orientations and 5 scales on the DAVIS dataset (resp. 16 orientations and 4 scales on VanHateren dataset). This choice of number of orientations and number of sacles maximizes prediction performance on the corresponding datasets.

**Polar Predictor**   We use 32 pairs of convolutional channels with filters of size $17 \times 17$ pixels, without biases (no additive constants) and no padding (ie. "valid" boundary condition). For the multiscale version (mPP), we use 16 pairs of convolutional channels with filters of size $11 \times 11$ pixels, without biases (no additive constants). The representation is computed inside a fixed Laplacian pyramid [10]. We used 4 scales for the DAVIS dataset (and respectively 3 scales for the VANH dataset). Within this multiscale representation, the learned filters are applied with zero padding (ie. "same" boundary condition).

**Quadratic Predictor**   For the multiscale version (mQP), we use the same analysis ($f_w$) and synthesis ($g_w$) hyperparameters as mPP. For the quadratic prediction mechanism, we use 16 groups of 4 convolutional filters (twice as many as for mPP). The quadratic predictor ($p_w$) operates on groups of 4 coefficients and contains 12 quadratic units. It is more expressive than the multiscale Polar Predictor architecture and contains phase advance as a special case.

**Vanilla CNN**   Finally, we implemented a more direct convolutional neural network predictor (**CNN**), that maps two successive observed frames to an estimate of the next frame [14]. For this, we used a CNN composed of 20 stages, each consisting of 64 channels, and computed with $3 \times 3$ filters without additive constants, followed by half-wave rectification (ie. ReLU). To facilitate learning, a skip connection copies the current frame $\mathbf{x}_t$ and the network only outputs residuals that get added to the current frame in order to predict the next frame: $\hat{\mathbf{x}}_{t+1} = \mathbf{x}_t + f_w([\mathbf{x}_t, \mathbf{x}_{t-1}])$. This model jointly transforms and processes pairs of frames to generate predictions, while both polar predictor (PP) and quadratic predictor (QP) separate spatial processing and temporal extrapolation.

**Unet**   The Unet architecture [15] is commonly used for image-to-image tasks. It has an analysis-synthesis structure with downsampling, upsampling, and skip connections between levels at the same resolution. We used a 5 levels architecture, each block consists of two convolutions, batch norm and rectification. The convolutions have filters of size $3 \times 3$, comprise 64 channels, and no additive bias. The end-to-end network comprises 20 non-linear stages with ReLU activations.

# D  Description of datasets and optimization

**DAVIS**  To train, test and compare these models, we use the DAVIS dataset [16], which was originally designed as a benchmark for video object segmentation. Image sequences in this dataset contain diverse motion of scenes and objects (eg., with fixed or moving camera, and objects moving at different speeds and directions), which make next frame prediction challenging. Each clip is sampled at 25 frames per second, and is approximately 3 seconds long. The set is subdivided into 60 training videos (4741 frames) and 30 test videos (2591 frames). We pre-processed the data, converting all frames to monochrome luminance values, and scaling their range to the interval $[-1, 1]$. Frames are cropped to a $256 \times 256$ central region, where most of the motion tends to occur, and then spatially down-sampled to $128 \times 128$ pixels.

**VanHateren**  We also consider a smaller video dataset consisting in footage of animals in the wild [48] which contains a variety of motions (animals in the scene, camera motion, etc.) and occlusions. The missing band at the top the frame is cropped, reducing the image size from $128 \times 128$ pixels to $112 \times 128$ pixels. The dataset is standardized to zero mean and unit variance, it is split into snippets of 11 frames, 292 snippets are used for training and 33 for testing. There is no spatial downsampling or whitening.

**Boundary handling**  The computation of this prediction error is restricted to the center of the image because moving content that enters from outside the video frame is inherently unpredictable. Specifically, we trim a 17-pixel strip from each side, yielding frames of size $94 \times 94$ pixels for the DAVIS dataset and $78 \times 94$ for the VanHateren dataset. Convolutions are computed with zero padding so that the outputs have the same size as inputs (the only exception is for the plain Polar Predictor, shown in Figure 6, where valid convolutions were performed).

**Training procedure**  We assume the temporal evolution of natural signals to be sufficiently and appropriately diverse for training, and do not apply any additional data augmentation procedures. We train on brief temporal segments containing 11 frames (which allows for prediction of 9 frames), and process these in batches of size 4. We train each model for 200 epochs on DAVIS using the Adam optimizer [49] with default parameters and a learning rate of $3 \cdot 10^{-4}$. The learning rate is automatically halved when the test loss plateaus. In the CNN, we use batch normalization before every half-wave rectification, rescaling by the standard deviation of channel coefficients (but with no additive bias terms).

# E  Additional results

**Additional dataset**  Note that the PSNR is the logarithm of the MSE in units of the signal: PSNR $= 10 \log_{10}(\text{MAX}^2/\text{MSE})$, where MAX is the maximum possible pixel value of the image. A PSNR of 0dB means that the squared peak signal and the mean squared error are equal; a PSNR of 10dB (resp. 20dB) means that the squared peak signal is 10 times (resp. 100 times) bigger than the MSE.

**Table 2: Performance comparison.** Prediction error computed on the VanHateren dataset. Values indicate mean (standard deviation) of Peak Signal to Noise Ratio (PSNR in dB) computed over 10 random seeds.

| | | | Method | | | | |
|---|---|---|---|---|---|---|---|
| split | Copy | cMC | SPyr | **mPP** | **mQP** | CNN | Unet |
| train | 27.10 | 28.56 | 29.50 | 30.16 (0.02) | 30.28 (0.04) | 31.16 (0.10) | **31.52** (0.42) |
| test | 26.41 | 27.93 | 28.83 | 29.11 (0.01) | 29.07 (0.04) | 29.09 (0.05) | **29.26** (0.06) |

**Computational costs**  The polar predictor described in this paper is lightweight: it runs rapidly and contains few parameters (two orders of magnitude less than CNN and Unet). The polar predictor is designed as an online method that could be applied to streaming data. Parameter counts, as well as training and inference time are reported in Table 3.

Notice that the Unet is also very efficient: it runs fast (most of the computation is applied to spatially downsampled coefficients). Note that the Quadratic Predictor (mQP) is slower. This model was

**Table 3: Number of trainable parameters and run times.** Mean training time (standard deviation) for 200 epochs on the VanHateren dataset (mean in min:sec, std in sec). Mean inference time (standard deviation) for a single video snippet containing 11 frames of size $112 \times 128$ pixels (mean in $ms$, std in $\mu s$). Training and inference time are computed on a NVIDIA A100 GPU.

| dataset | Method | | | |
|---|---|---|---|---|
| | **mPP** | **mQP** | CNN | Unet |
| # parameters | **7,744** | 15,776 | 666,496 | 591,041 |
| training time | **10:24** (20) | 46:14 (29) | 32:17 (50) | 11:14 (17) |
| inference time | 9.71 (11) | 19.8 (8) | 9.57 (8.4) | **2.7** (8.2) |

developed to suggest a connection with physiology (each element recapitulates know functional building blocks of primate visual physiology), not to improve performance or efficiency.

**Decoupling analysis and synthesis**    When decoupling the analysis and synthesis transforms (ie. untying the weights), a Polar Predictor (not multiscale) achieved a similar prediction performance on the VanHateren dataset (tied: train 29.87/ test 28.83; vs. untied: train 29.99 / test 28.87 dB). The projection and basis vectors (ie. the filters in the analysis and synthesis transforms) align as training progresses.

