# A  Description of architectures

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

**Figure 6:** Filters of a polar predictor trained to predict natural videos from the DAVIS dataset. The 32 pairs of convolutional filters are sorted by their norm and their amplitude spectrum is displayed at corresponding locations on the right panel. Observe that the filters are selective for orientation and spatial frequency, tile the frequency spectrum, and form quadrature pairs.

| $x(t-1)$ | $x(t)$ | $x(t+1)$ | cMC | SPyr | **mPP** | **mQP** | CNN |
|---|---|---|---|---|---|---|---|

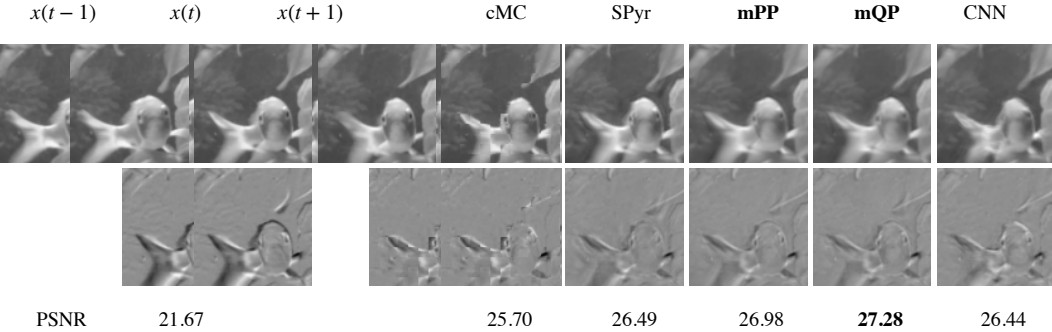

| PSNR | 21.67 | | 25.70 | 26.49 | 26.98 | **27.28** | 26.44 |

**Figure 7:** A typical example image sequence from the DAVIS test set. The first three frames on the top row display the unprocessed images, and last five frames show the respective prediction for each method (with their shorthand above). The bottom row displays error maps computed as the difference between the target image $x(t+1)$ and each predicted next frame on the corresponding position in the first row. Images, predictions and error maps are all shown on the same scale.