# OpenReview forum: "A polar prediction model for learning to represent visual transformations"
_NeurIPS.cc/2023/Conference — NeurIPS 2023 poster_

### Official Review · Reviewer_7L6B · 2023-07-06

**Soundness:** 3 good
**Presentation:** 3 good
**Contribution:** 3 good
**Rating:** 7
**Confidence:** 4

**Summary:**

The authors create self-supervised video prediction model that relies on principles of the Fourier shift theorem. The approach is inspired by work suggesting that humans rely on perceptual straightened representations to support prediction. The authors first validate their method on toy cases of simple translation and rotation and later show that their model outperforms motion compensation and a some standard deep learning learning models at next frame prediction. Finally, they draw connections between their model and early visual processing in biological vision and discuss their model's impact more widely to computer vision (a simpler and more principled way to get natural video representations).

**Strengths:**

The methods are well thought-out and of good quality. The connection between perceptual straightening and the Fourier shift theorem is clever and novel. The use of both local and multi-scale processing in the model's design is is thoughtful and clear grounded in principles of human visual processing. In general, the authors make it very clear how their method relates to prior work -- this makes it the novelty and significance of the work clear.

The analyses of the model are good. I appreciate that the authors test their approach in synthetic data and then move to larger natural video datasets like DAVIS. Some good baselines are included in the analysis (copying frames, basic CNN), and there is an analysis of the filters learned by the model.

**Weaknesses:**

The main weakness I see not having enough baseline comparisons to existing methods in video prediction or video representation learning. The work only compares their model to a basic CNN which makes it difficult to understand how their model compares to more state of the art approaches or even older methods grounded in neuroscience like PredNet. Reporting scores for more models or adding another dataset (like Kinetics) with more performance measures like mse or lpips would address this point. In addition, more detail on compute and training requirements would help with understanding the significance of their approach.

**Questions:**

I generally found the paper clear and interesting, but I would like to hear more on the following points:

Could the authors discuss how their model might compare new architectures ViT and diffusion models for video prediction? Could the authors report MSE, SSIM, and lpips scores to help me get a better sense of how their model compares to other approaches? To that end, do the authors have any performance results on other datasets that are used in video prediction (for example moving MNIST or Kinetics)?

Do the authors have any thoughts on other downstream tasks their video representation learning framework could be useful for? Or are there even aspects of video prediction that your model might do better at being grounded in a straightening representation space (for example long range prediction vs short range prediction)?


**Limitations:**

Limitations are not widely discussed, but aside from the weaknesses I highlighted I do not see a glaring limitation that needs to be addressed.

---

> ### Author Rebuttal · Authors · 2023-08-10
>
> Thank you for your review and comments.
>
> * **Architectures, metrics and datasets**: we have added a Unet architecture to our comparisons, see global response. We could not find a reference PyTorch implementation of the PredNet architecture, but to ease comparison in the future, we intend to release our code upon acceptance. Since a main motivation of our study was to develop an interpretable prediction method, we did not consider architectures like ViT or Diffusion models, although we do expect that such expressive architectures would perform very well (given enough data). We also computed SSIM and observed that the results come out similar and do not change our initial interpretation, see global response. We have also applied our method to the UCF-101 dataset and again the trend in the results was similar and chose not to include it to reduce clutter. We are currently designing toy datasets to tests specific hypothesis regarding video prediction (especially at occlusion boundaries).
> * **Downstream**: see applications in global response.
> * **Limitations**: see global response.

---

> > ### Comment · Reviewer_7L6B · 2023-08-19
> > **response to rebuttal**
> >
> > Thanks you for adding the additional work on architecture, metrics, and datasets. The paper is more solid with these additions, but the essential contribution is the same. I still think the paper is solid and deserves to be accepted, so I keep my score.

---

### Official Review · Reviewer_nLAh · 2023-07-07

**Soundness:** 2 fair
**Presentation:** 2 fair
**Contribution:** 2 fair
**Rating:** 6
**Confidence:** 2

**Summary:**

Motivated by Fourier shift theorem and its group-theoretic generalization, this paper proposed a new video prediction model.

**Strengths:**

1. Biological modeling: computational elements can be used to describe primate V1 responses.
2. Fourier shift theorem and its group-theoretic generalization were incorporated into future prediction

**Weaknesses:**

1. Evaluation: only simple baselines like CNN were included for comparison. Only one quantitative metric, PSNR, was used. Other metrics like SSIM or FVD can be more informative. No demonstration of any qualitative results.
2. Loss: this framework used MSE as loss for optimization, which indicated this setting didn't consider any uncertainty factors.

**Questions:**

I am not familiar with computational neuroscience. Therefore, I can only provide useful feedback with respect to generative performance.

**Limitations:**

No limitations were discussed in the paper.

---

> ### Author Rebuttal · Authors · 2023-08-10
>
> Thank you for your review and comments.
>
> * **Evaluation**: we computed SSIM, the results come out similar and do not change our initial interpretation, see global response. A qualitative example and its interpretation is included in Figure 7 of the supplementary material in the original submission. We intend to move it to the main text and to add more examples in the appendix.
> * **Loss:** we used MSE for convenience because it is standard and amenable to optimization via gradient descent - although it is well known that image signals are not well captured by a gaussian model. If you have any suggestion, I would be curious to know what you are thinking about.
> * **Computational neuroscience:** one of the main contributions of this paper is to connect predictive video processing with modeling of the early visual system in computational neuroscience. By starting from fundamentals (Fourier shift theorem) and considering an abstract formulation (learning the representation of group transformations), we have exposed the unity of these two approaches. In particular, normative models of V1 receptive fields are typically based on coding efficiency and sparse coding, while our approach relies on the prediction principle instead and naturally accounts for several well documented phenomena such as normalized simple cell responses and direction selective complex cell responses.

---

> > ### Comment · Reviewer_nLAh · 2023-08-17
> > **Response to rebuttal**
> >
> > Most of my concerns are addressed therefore I increase my rating to weak accept.
> > For loss, if you want to add randomness into your prediction system, you can try to use ELBO loss.

---

### Official Review · Reviewer_i4iN · 2023-07-07

**Soundness:** 3 good
**Presentation:** 3 good
**Contribution:** 3 good
**Rating:** 6
**Confidence:** 3

**Summary:**

This research develops a self-supervised representation-learning framework that uses the regularities of natural videos to make accurate predictions. The architecture is inspired by the Fourier shift theorem and trained for next-frame prediction. The approach can discover irreducible representations of smooth commutative groups in data. It achieves similar performance comparing to the traditional motion compensation and conventional deep networks, while being interpretable and fast. The framework is implemented using normalized simple and direction-selective complex cell-like units, similar to primate V1 responses, demonstrating its potential to explain how the visual system makes temporal predictions.


**Strengths:**

-	The paper leverages the temporal regularity of the natural video and propose an video compression algorithm for temporal linearization and compression. The model exploits the local symmetries present in the temporal evolution of images therefore save computational resource comparing to the end-to-end CNN models. This is novel and original for video compression and image representation learning.
-	The paper suggests an important research direction: instead of producing an end-to-end generic deep learning framework, the authors leverages the temporal regularity of the natural image sequence to reduce the computational cost. Human mind utilizes certain evolutionary bias to improve its efficiency, so should we exploit these shortcut and apply them into the computer vision research.
-	The paper’s method is elaborated clearly with precise mathematical notation. The presentation flow is clear although it would be benefit to draw illustrative figures to show intuitively show the main idea of the paper.


**Weaknesses:**

-	Despite the promising direction of research, the comparison between CNN models and the proposed method doesn’t demonstrate a significant improvement in terms of numbers. Further error bar would provide more concrete idea of how much the method has improved.
-	Other evaluation metrics could be useful such as the time of performing compression, as well as the computational resource. It would still be a good contribution if the model can achieve the same performance but using significantly less resources. Although the paper briefly mentioned this, it would be more thorough to outline the resource saving using the proposed method.
-	A description of the overall algorithm would make the presentation more clear.


**Questions:**

-	Detailed experimental setup and dataset description are missing in the main text.
-	Would be beneficial to see other evaluation metrics on compression decoding quality and efficiency would be beneficial other than only PSNR metrics, e.g. bitrate, inference time etc.
-	Will the learned representation also be useful for image classification or other static tasks?



**Limitations:**

The authors have drawn parallels between the computational steps of this approach and classical models of early visual processing in primates. However, the evidence supporting these parallels could be better substantiated. Quantitative comparison to physiological data may not be a straightforward task, and potential challenges in doing so should be discussed.

---

> ### Author Rebuttal · Authors · 2023-08-10
>
> Thank you for your review and comments.
>
> * **Compression**: The prediction method presented in this paper does not constitute a full video compression engine. Indeed the error are not quantized and we have not considered a full rate-distortion tradeoff. But we envision a possible application of the ideas developed in this study to coding because prediction is a key step in video coding.
> * **Error bars**: we added standard deviation of prediction error over multiple runs which helps interpret the relative performance of the algorithms, see Table 1 in rebuttal pdf.
> * **Costs**: our claim is that the performance of the proposed methods is on par with that of more complex models while using significantly fewer parameters and remaining interpretable, see Table 2 in rebuttal pdf.
> * **Algorithm description**: we added equations to describe the polar predictor algorithm, using concise vector notation, see rebuttal pdf.
> * **Detailed setup**: we omitted implementation details from main text due to space limitations, this information is available in the supplementary material of the original submission and we have improved the readability of the corresponding section.
> * **Metrics**: we computed SSIM, the results come out similar and do not change our initial interpretation, see global response.
> * **Downstream task**: we anticipate that segmentation, rather than recognition, would potentially benefit from the representation learned on the prediction task, a more detailed study is left for future work, see global response.
> * **Comparing with physiology**: We agree that quantitative comparison with physiological data is often challenging, especially given the amount of noise and the limited number of trials. In this study we operationalized a construction of visual signal prediction that can be cast in the same modeling framework that has been used to describe neural responses - and we aim to apply it to data in a future study.

---

### Official Review · Reviewer_HG1W · 2023-07-09

**Soundness:** 4 excellent
**Presentation:** 3 good
**Contribution:** 3 good
**Rating:** 7
**Confidence:** 3

**Summary:**

The authors in this work propose a new self-supervised learning technique that is aimed to perform predictive processing of natural videos (although the approach seems broad enough to be applicable to other sequential signals with similar inductive priors to vision). The authors develop two parameter-efficient architectures (multiscale quadratic and polar prediction mechanisms, mQP, mPP) based on Fourier shift theorem and its group theoretic generalization and train them to perform next-frame prediction from natural videos. Experimental results show that their proposed abovementioned mQP and mPP architectures outperform relevant conventional motion coding baselines and a supervised CNN on video reconstruction on the two video datasets of VanHateren and DAVIS. It is very intriguing that an interpretation of the learned filters in their architecture shows similarity to canonical receptive fields observed in V1 cells, hence providing a new theoretical hypothesis for a potential function of V1 cells in performing future prediction.

**Strengths:**

+ The proposed work seems theoretically very sound, the proposed architectures mQP and mPP are designed ground up based on the extension of the Fourier shift theorem to account for transformations that occur in natural videos. Adequate mathematical and illustrative derivation of the proposed algorithm is very useful to grasp the underlying mechanism even for non-experts in this specific area such as myself.
+ It is super interesting that canonical receptive field structure identified in V1 cells is emergent from the proposed self-supervised architecture and this serves as a foundation for the hypothesis of V1 functioning in temporal straightening and future prediction.
+ Experimental results show that the proposed approach is clearly well-performing in terms of future frame prediction. These evaluations aren't very extensive like typical machine learning papers but the message from the paper is clearly conveyed in this demonstration. Additionally the link to biological modeling that I mentioned previously makes the paper a fresh read with many interesting avenues.
+ The writing is great overall and related prior work has been covered quite comprehensively.

**Weaknesses:**

- In the experimental evaluation, it maybe good to include the number of parameters and runtime (training and inference if possible, but certainly for inference) for each algorithm. I believe this will provide a more full picture and show how much more efficient the proposed architecture is.
- It would be great if the authors also add measures of variance for the performance of various algorithms on the two datasets compared in the evaluation section.
- Although the authors have briefly touched upon potential alternate usecases of the proposed algorithm in other vision tasks (segmentation, direction estimation etc.) it would be good for readers in the broader audience to elaborate further on this.
- Limitations of the proposed work aren't discussed in much detail, I only see that one potential limitation listed here is the inability of current proposed algorithms to generalize to non-commutative groups which is left for future scope. I encourage the authors to further discuss this in more detail and add other directions they see for extension of this work to the paper.

**Questions:**

- Please see the weaknesses section above for my suggestions.

**Limitations:**

The authors have addressed limitations of this work.

---

> ### Author Rebuttal · Authors · 2023-08-10
>
> Thank you for your review and comments.
>
> * **Costs**: see global response and Table 2 of rebuttal pdf.
> * **Error bars**: see global response and Table 1 of rebuttal pdf.
> * **Applications**: see global response.
> * **Limitations**: see global response.

---

### Official Review · Reviewer_Zhuh · 2023-07-25

**Soundness:** 3 good
**Presentation:** 3 good
**Contribution:** 3 good
**Rating:** 6
**Confidence:** 3

**Summary:**

The authors present a self-supervised representation-learning framework inspired from the idea of continuous deformations of objects in videos. The framework allows for next-frame prediction given previous ones, and borrows ideas from the Fourier shift theorem. The framework is composed of three stages - an analysis transformation $f_w$ of given frames, a prediction transformation $p_w$ that predicts the next frame representation given previous ones, and a synthesis transformation $g_w$ which transform the latent representation to the actual frame. $g_w$ is implemented as the inverse of $f_w$, and the two share parameters. The authors present two variants of their framework, the polar predictor (PP) where $p_w$ is a fixed polar extrapolation, and a learnt $p_w$ variant termed the quadratic predictor (QP). The authors also detail results on next-frame prediction task, and show that their method surpasses baseline techniques.

**Strengths:**

- The idea of exploiting the temporal symmetries and local deformations in videos to derive a natural framework for next-step prediction is interesting and has the potential to provide interpretability for sequence prediction tasks.

- The idea of sharing parameters between the synthesis and analysis operators, contributes clarity to the framework's underlying process.

- Multiscale processing using a Laplacian pyramid is an interesting solution to overcome video transformations of different scales.

- The proposed framework shows promising results in the provided evaluation.

**Weaknesses:**

- Technical details regarding the implementation of the learned transformations is missing from the main paper.

- The section on multiscale processing using a Laplacian pyramid, in my opinion, is hard to follow. A figure illustrating the process would greatly help.

- The paper could benefit from some ablation tests, for example - using shared weights for $f_w$ and $g_w$ as opposed to separate weights, using different network architectures/depths and so on.

- The authors present the method as a representation-learning framework, but it's not clear how the learnt representations could assist with tasks beyond next-frame prediction? I would appreciate listing some additional possible applications. Video compression is briefly mentioned as a possibility, but no details are given (are the representations compact relative to the original frame?)

- Regarding the comparison to baseline methods - the baseline CNN architecture chosen for comparison seems quite arbitrary. On the one hand, the authors claim their method achieves comparable or better results using less parameters than the CNN, but on the other hand - no baseline CNN with the same number of parameters is given for comparison. In addition, the architecture seems arbitrary - 20 layers with the same number of channels, no skip connections in between (only one that wraps the entire network so that the predictions are residuals) and no information regarding non-linearities and/or normalization layers. It seems to me that choosing an of-the-shelf architecture for general image-to-image tasks (and slightly modifying the input/output) would have been a much better and competitive baseline. In addition, the realted work section mentions additional works on next-frame prediction (for example using LSTMs), which could contribute to the comparison. I would appreciate the author's response on this.

**Questions:**

- Have you tried using the proposed framework to predict additional frames, as opposed to only the next one? It seems quite intuitive and straightforward using the proposed method since it only requires an additional step in the polar extrapolation.

- Are there any latency constraints to the proposed approach? For example in the transformation to polar coordinates.

**Limitations:**

After reading through the paper a few times, I couldn't find any addressed limitations (although I can't point to any unaddressed limitations).

---

> ### Author Rebuttal · Authors · 2023-08-10
>
> Thank you for your review and comments.
>
> * **Implementation details**: the learned transformations and their rationale are introduced in the main text; but, due to space limitation, their full description (architectures, datasets and optimization) is relegated to the supplementary material. We improved the readability of these sections, and in the rebuttal pdf, we included equations to summarize the polar prediction method.
> * **Laplacian pyramid**: we drew a multiscale diagram and included it as a figure in the rebuttal pdf, we also clarified the notation and text in the corresponding caption. The multiscale processing in mPP and mQP can be thought of as a preprocessing step: the analysis ($f_w$), prediction ($p_w$) and synthesis ($g_w$) steps are applied to the Laplacian coefficients ($\Delta \mathbf{x}$) instead of the image itself ($\mathbf{x}$).
> * **Ablations**: we have explored decoupling the analysis and synthesis transformations (ie. untying feedforward and feedback weights) of a Polar Predictor (not multiscale). It achieved a similar prediction performance on the VanHateren dataset (tied: train 29.87/ test 28.83; vs. untied: train 29.99 / test 28.87 dB) and observed that the learned projection and basis vectors align as training progresses.
> * **Applications**: compression and segmentation are two downstream tasks where the proposed method may prove effective, see global response for a brief discussion.
> * **Comparison methods**: Although, we have not systematically optimized the hyperparameters of the CNN (we are reusing the dnCNN [1] architecture which has proven successful in other image-to-image tasks), we have added a U-net baseline - see global response. We have not studied fully recurrent architectures (such as LSTMs).
> * **Beyond next-frame**: We have observed that, as expected, the quality of the prediction quickly degrades with the temporal horizon, but we have not carefully compared the performance decay rate of different prediction methods. Computing further predictions in the latent space and then generating the corresponding frames using the synthesis transform is an interesting suggestion. Now that you mention it, it occurs to us that such predictions could be compared to a recursive approach where the next frame is predicted and then fed back in as an input - we intend to run these experiments.
> * **Speed**: For parameter count and latency, see global response and Table 2 in the rebuttals pdf.
>
> [1] Zhang, K., Zuo, W., Chen, Y., Meng, D. and Zhang, L., 2017. Beyond a gaussian denoiser: Residual learning of deep cnn for image denoising. IEEE transactions on image processing

---

> > ### Comment · Reviewer_Zhuh · 2023-08-12
> > **Response to rebuttal**
> >
> > I have read the other reviewers' comments and the authors' responses.
> >
> > I believe the rebuttal properly answered some of the concerns raised by myself and the fellow reviewers. Specifically, I appreciated the clarifications, added diagram and additional baseline (U-Net).
> >
> > I have updated the score accordingly. The reason why I chose not to give a higher score, mainly lies in the currently undeveloped applications to the representation learning framework, and the fact that the U-Net baseline seems to be comparable in terms of accuracy, and superior in terms of latency, despite its simplicity (although the reduced number of parameters is comforting).

---

### Author Rebuttal · Authors · 2023-08-10

We thank the reviewers for their comments and questions.
The points that were raised by multiple reviewers are addressed in this global response
and the other questions are addressed in individual responses.

**Additions/extensions:**
* **Error bars**: single run prediction errors were reported in the initial submission, we now include average prediction error (and standard deviation) computed over ten random seeds - the results are assembled in Table 1 of the rebuttal pdf. Notice that the methods we intoduced (mPP and mQP) outperform simple baselines (Copy, cMC and SPyr) and rival more complex architectures (CNN and Unet) while being less variable (and also more interpretable).
* **New comparison algorithm**: we trained a Unet [1] for video prediction, it is a standard architecture for image-to-image tasks like segmentation. It has an encoder-decoder like structure with downsampling, upsampling, and skip connections between levels at the same resolution. We used a 5 levels architecture, each block consists of two convolutions, batch norm and rectification. The convolutions have filters of size 3 by 3, comprise 64 channels, and no additive bias. As expected, this architecture is very efficient: it runs fast (most of the computation is applied to spatially downsampled coefficients), it is the most performant on the VanHateren dataset, but it overfits on DAVIS (even though we limited the size of the model by using only 64 channels). This architecture comprises 20 non-linear stages (ReLU), which is to be compared to the single polar non-linearity in the polar predictor (mPP).
* **Costs**: The polar predictor described in this paper is lightweight: it runs very fast and contains few parameters (two orders of magnitude less than CNN and Unet). The polar predictor is designed as an online method that could be applied to streaming data and it is indeed very quick to train and to run. Parameter counts, as well as training and inference time are reported in Table 2 of the rebuttal pdf.
    * This project was developed on a NVIDIA A100 GPU, and used small datasets: 3575 frames of size 112 by 128 for VanHateren, 7332 frames of size 128 by 128 for DAVIS.
    * Note that the Quadratic Predictor (mQP) is slower, indeed we used a naive implementation of this model as it is not intended for performance but to build a connection with the computational neuroscience literature by showing how each computational element recapitulates know functional building blocks of primate V1 physiology.
* **Performance metric**: We report performance measured by PSNR (which is the logarithm of MSE in units of the signal). We also computed SSIM and observed that it did not change the trend or the interpretation of the results. A note on interpreting PSNR results: a PSNR of 0dB means that the signal and the error have the same variance, a PSNR of 20dB (resp. 40dB) means that the signal variance is 10 times (resp. 100 times) bigger than the error variance.

**Writing improvements:**
* **Unify algorithm**: Each element of the prediction computation and loss function were introduced gradually through the development of main text. To enhance clarity, we gather the equations that precisely describe the architecture and objective, using compact vector notation. These equations are at the top of rebuttal pdf.
* **Applications**: We briefly outline potential use of the learned mPP representation for other tasks.
    * Compression: in typical video compression engines like MPEG, the encoder computes and transmits motion vectors (slow to compute for the encoder but fast to apply for the decoder) as well as correction bits. Since our polar predictor can very rapidly compute a predicted frame, we envision a compression method that only sends prediction errors - thereby potentially saving substantial bitrate on motion vector. In practice, the feasibility of this approach would depend on the propagation of error quantization. Actually testing these ideas at different quantization levels is a large endeavor that is well beyond the scope of the present study.
    * Segmentation: visual content that moves together might belong to the same object, therefore the equivariant part of the polar predictor could be used to infer object boundaries by clustering phases changes. We expect that heading direction estimation could be approach in a similar way and leave this.
* **Limitations**: Throughout the main text, we have formulated assumptions and expressed the limitations of our prediction methods. We will consolidate them into a paragraph to be included in the discussion section of the main paper.
   * Our method has several limitations. The polar and quadratic prediction mechanisms assume fixed amplitude and linear phase increments in successive frames. Therefore these parameterizations cannot capture acceleration or longer temporal dependencies, and they also cannot precisely handle fast and long range displacements or deformations. In practice, and as can be seen in the qualitative prediction examples, error often occur at occlusion boundaries. We rely on a simple MSE objective function which corresponds to a simplistic gaussian noise model on the pixels, which we know to be an inadequate description of visual signals. The synthetic translation and rotation examples show that different representations are required to capture different group transformations, but the polar predictor architecture is forced to find a compromise between the two. We are now instead considering representations that can modulate their gain in order to adapt to the transformation currently acting in the data. Finally, recall that our choice of parameterization was motivated by the representation theory of commutative Lie groups, it can therefore not handle non-Abelian transformations and we leave exploration of QP mechanism's expressivity for future work.

[1] Ronneberger, Fischer, Brox 2015 U-net: Convolutional networks for biomedical image segmentation. In MICCAI

---

> ### Author Response · Authors · 2023-08-21
> **additional sections in supplementary material**
>
> We thank the reviewers for engaging with our rebuttal and would like to mention sections added to the supplementary material:
> * additional metrics: we also report the MSE and SSIM values for our experiments (on top of the PSNR in main text)
> * a new section on comparing Polar Prediction with related methods and objective functions. In the original manuscript, these methods were mentioned in the related works section of the main text, now the precise formulation of the following methods is reviewed:
> Slow Feature Analysis (Wiskott, Sejnowski),
> Dynamics Mode Decomposition (Mezic, Schmid, etal),
> Sparse Transformational Invariants (Cadieu, Olshausen),
> Phase-based Video Motion Processing (Wadhwa etal),
> Toroidal Subgroup Analysis (Cohen, Welling),
> Sparse Manifold Transform (Chen etal).
> * additional qualitative predictions examples to let the reader visually compare the methods considered in this paper.

---

### Decision · Program_Chairs · 2023-09-21

**Decision:**

Accept (poster)

**Comment:**

The paper proposes a new self-supervised learning technique for predictive processing of natural videos. The reviewers agree that the the ideas are novel and elegant. Several reviewers raised concerns about insufficient evaluation, which the authors addressed by adding another baseline and reporting that their results also hold with a different evaluation metric (SSIM; however, no numbers presented). Although the paper could probably be further improved, it appears to make an innovative contribution already in its current form.